# Novel Indane Derivatives with Antioxidant Activity from the Roots of *Anisodus tanguticus*

**DOI:** 10.3390/molecules28031493

**Published:** 2023-02-03

**Authors:** Chun-Wang Meng, Hao-Yu Zhao, Huan Zhu, Cheng Peng, Qin-Mei Zhou, Liang Xiong

**Affiliations:** 1State Key Laboratory of Southwestern Chinese Medicine Resources, School of Pharmacy, Chengdu University of Traditional Chinese Medicine, Chengdu 611137, China; 2Institute of Innovative Medicine Ingredients of Southwest Specialty Medicinal Materials, School of Pharmacy, Chengdu University of Traditional Chinese Medicine, Chengdu 611137, China; 3Innovative Institute of Chinese Medicine and Pharmacy, Chengdu University of Traditional Chinese Medicine, Chengdu 611137, China

**Keywords:** *Anisodus tanguticus*, indanes, indenofuran, absolute configuration, antioxidant activity

## Abstract

Four novel indane derivatives, anisotindans A–D (**1**–**4**), were isolated from the roots of *Anisodus tanguticus*. Their structures were established using comprehensive spectroscopic analyses, and their absolute configurations were determined by electronic circular dichroism (ECD) calculations and single-crystal X-ray diffraction analyses. Anisotindans C and D (**3** and **4**) are two unusual indenofuran analogs. ABTS^•+^ and DPPH^•+^ assays of radical scavenging activity reveal that all compounds (**1**–**4**) are active. Specifically, the ABTS^•+^ assay results show that anisotindan A (**1**) exhibits the best antioxidant activity with an IC_50_ value of 15.62 ± 1.85 μM (vitamin C, IC_50_ = 22.54 ± 5.18 μM).

## 1. Introduction

*Anisodus tanguticus* (Maxim.) Pascher is a folk medicine commonly used in northwest and southwest China [1,2]. The roots of *A. tanguticus* can relieve pain and spasms, promote blood circulation, remove blood stasis, stop bleeding, and strengthen muscles. Moreover, they are often used in the clinical treatment of pain, ulcer, colitis, gallstone, traumatic injury, catagma, hemorrhage, anesthesia, and motion sickness [3,4,5]. Due to its scarce plant resources, wild *A. tanguticus* was once listed as a class II protected endangered plant in the List of National Key Protected Wild Plants (the first batch). However, the implementation of long-term environmental protection strategies and the establishment of several planting bases in Ganzi and Aba (Sichuan Province) have significantly improved the plant resources of this species over the past years. Today, *A. tanguticus* is considered an important economic plant due to its high content of tropane-type alkaloids. Moreover, this plant is currently the natural resource of anisodamine, anisodine, hyoscine, and cuscohygrine.

In light of their significant biological activities, such as anti-shock effect [6], amelioration of hypoxic injury [7], cardioprotective effect [8], and alleviation of angina symptoms [9], the tropane-type alkaloids in *A. tanguticus* have been extensively researched. However, few studies are available on the non-alkaloid components of the plant. Previously, we had isolated a series of compounds from *A. tanguticus*, including four new sesquiterpenoids with an unprecedented skeleton [10,11]. In this study, four novel indane derivatives (**1**–**4**) were isolated (Figure 1). Indanes are generally characterized by anti-tumor [12], anti-microbial [13,14,15], anti-inflammatory [16,17], neuroprotective [18], and antioxidant activities [19], and, thus, they have been used to develop various drugs, such as indacaterol, aprindine, and donepezil. Knowing that indanes have significant free radical scavenging activity [19], the scavenging activity of compounds **1**–**4** is analyzed herein using ABTS^•+^ and DPPH^•+^ assays.

## 2. Results

### 2.1. Structure Elucidation

Anisotindan A (**1**) was obtained as colorless crystals, and based on HR-ESI-MS analysis, its molecular formula is C_13_H_18_O_3_, with five degrees of unsaturation (*m*/*z* 245.1157, calcd. for C_13_H_18_O_3_Na, 245.1154). The ^1^H NMR spectrum of **1** displays signals corresponding to *ortho*-coupled aromatic protons [*δ*_H_ 6.77 (1H, d, *J* = 8.4 Hz) and 6.59 (1H, d, *J* = 8.4 Hz)], an oxymethylene group [*δ*_H_ 3.48 (1H, dd, *J* = 10.8, 6.0 Hz) and 3.45 (1H, dd, *J* = 10.8, 6.0 Hz)], two aliphatic methylene groups [*δ*_H_ 2.91 (1H, dd, *J* = 15.0, 9.0 Hz), 2.76 (1H, dd, *J* = 15.0, 9.0 Hz), and 2.82 (2H, m)], one aliphatic methine group [*δ*_H_ 2.65 (1H, m)], two tertiary methyl groups [*δ*_H_ 2.09 (3H, s) and 1.19 (3H, s)], and three exchangeable protons [*δ*_H_ 7.77 (1H, s), 3.70 (1H, t, *J* = 6.0 Hz), and 3.28 (1H, s)]. The ^13^C NMR and DEPT spectra exhibit carbon resonance signals that can be assigned to the protonated units listed above, as well as to five quaternary carbons (*δ*_C_ 154.4, 144.3, 134.7, 120.4, and 73.7). Comprehensive 2D NMR analysis reveals ^1^H-^1^H COSY correlations of H_2_-1/H-2/H_2_-3 and H-6/H-7, in conjunction with HMBC correlations of H_2_-1 with C-3, C-3a, C-7, and C-7a, and of H_2_-3 with C-3a and C-4 in the indane framework of **1** (Figure 2). Based on the HMBC correlations from H_2_-1 and H_2_-3 to C-2′; from H-2 to C-1′, C-2′, and C-3′; from H_2_-1′ to C-2, C-2′, and C-3′; from H_3_-3′ to C-2, C-1′, and C-2′; from OH-2′ to C-2, C-2′, and C-3′; and from OH-1′ to C-1′ and C-2′, as well as the ^1^H-^1^H COSY correlation of H_2_-1′/OH-1′, a 1,2-dihydroxyisopropyl unit is established at C-2. The HMBC correlations of H_3_-4′ with C-3a, C-4, and C-5, and of OH-5 with C-4, C-5, and C-6 indicate that a methyl group and a hydroxy group are substituted at C-4 and C-5, respectively. Finally, single-crystal X-ray crystallography analysis [Flack parameter = 0.02(13)] reveals that the absolute configuration of **1** is 2*R*,2′*S* (Figure 3).

Anisotindan B (**2**) was obtained as a white powder, and its molecular formula was found to be C_13_H_18_O_3_, the same as **1**. As shown in Table 1, the ^1^H and ^13^C NMR data of **2** are highly similar to those of **1**, which suggests that the former might be an epimer of the latter. This is further confirmed by 2D NMR analysis (HSQC, ^1^H-^1^H COSY, and HMBC). Thus, compound **2** may be identified as (2*S*,2′*S*)-**2** or its enantiomer. A comparison of the calculated and experimental ECD data (Figure 4) reveals that the absolute configuration of compound **2** is 2*R*,2′*R*.

Anisotindan C (**3**) was obtained as colorless crystals, and it has the molecular formula C_13_H_16_O_3_, with six degrees of unsaturation (two fewer protons than compounds **1** and **2**), as evidenced by HR-ESI-MS analysis (*m*/*z* 243.0996, calcd. for C_13_H_16_O_3_Na, 243.0997). The ^1^H and ^13^C NMR spectra of compound **3** suggest that this compound is an analog of compound **2**. Indeed, comparison of the NMR data corresponding to the two compounds reveals that a methylene group in **2** is replaced by an oxymethine group [*δ*_H_ 5.47 (1H, d, *J* = 6.3 Hz), *δ*_C_ 88.1] in compound **3**. In addition, the signal of the exchangeable hydrogen proton (OH-1′) is observed in the ^1^H NMR spectrum of compound **2**, but not in the spectrum of **3**. This indicates that compound **3** is an ether derivative of compound **2**. The 2D NMR spectra of compound **3** confirmed its planar structure, especially the HMBC correlation of H-8b to C-2 (Figure 2). Moreover, H-3a, H-8b, and OH-3 in **3** have the same orientation, as evidenced by the enhancement of H-3a and OH-3 signals upon the irradiation of H-8b in 1D NOE spectroscopy analysis. The small coupling constant between H-3a and H-8b (*J*_3a,8b_ = 6.3 Hz) also indicates that H-3a and H-8b are *cis* oriented [20]. Single-crystal X-ray diffraction analysis [Flack coefficient 0.08(7)] (Figure 3) shows that the absolute configuration of **3** is 3*R*,3a*S*,8b*S*.

Anisotindan D (**4**) is an isomer of compound **3**, as indicated by HR-ESI-MS, ^1^H, and ^13^C NMR data. Comparison of the ^1^H and ^13^C NMR spectra of the two compounds reveals that the compound **4** is an isomer of compound **3**. As shown in Figure 2, the ^1^H-^1^H COSY and HMBC correlations reveal the presence of the 3,3a,4,8b-tetrahydro-2*H*-indeno [1,2-*b*]furan skeleton. In addition, the HMBC correlations of OH-3 with C-2, C-3, and C-1′; OH-7 with C-6, C-7, and C-8; and of H_3_-2′ with C-7, C-8, and C-8a confirm that two hydroxy groups and a methyl group are substituted at C-3, C-7, and C-8, respectively. Based on the NOE correlations of H-8b with H-3a and OH-3, as well as the small coupling constant between H-3a and H-8b (*J*_3a,8b_ = 6.3 Hz), H-3a, H-8b, and OH-3 in **4** have the same orientation. As shown in Figure 4, the calculated ECD spectrum of (3*S*,3a*R*,8b*R*)-**4** is consistent with the experimental spectrum, and thus, the absolute configuration of compound **4** is 3*S*,3a*R*,8b*R*.

### 2.2. Antioxidant Activities

As shown in Table 2, compounds **1**, **2**, **3**, and **4** exhibit ABTS free radical scavenging activities, with IC_50_ values of 15.62 ± 1.85, 40.92 ± 7.02, 43.93 ± 9.35, and 32.38 ± 6.29 μM, respectively. The activity of compound **1**, the most potent scavenger, is even stronger than that of vitamin C (VC, IC_50_ = 22.54 ± 5.18 μM). Based on the DPPH^•+^ assay, compound **1** also has an antioxidant effect, with an IC_50_ value of 68.46 ± 17.34 μM. However, the remaining compounds do not exhibit antioxidant activity, even at concentrations as high as 100 μM. Interestingly, the antioxidant activities of epimers **1** and **2** are quite different, despite the similar structures of the two compounds (differ only in the absolute configuration of C-2′).

## 3. Discussion

Indanes are a class of small organic molecules with a benzocyclopentane skeleton that can be substituted with 4-aminobenzylidene, gallic acid, piperidine, cyclohexadienone, or nucleobase to form indane analogs with diverse structures and significant activity [21]. Many reports are available in the literature regarding the synthesis of indane analogs via Friedel–Crafts-type, Michael-type, and Heck-type cyclization reactions [21,22]. However, reports on naturally occurring indanes are scarce. Notably, *A. tanguticus* seems to contain several indane derivatives, including rare polyhydroxy indenofurans.

Oxidative stress, a condition induced by the excessive generation of free radicals, is considered to be an important cause of human disease and aging. Indeed, the accumulation of reactive species in cells can lead to DNA damage, as well as protein and lipid degradation, which affects normal physiological functions [23]. Therefore, antioxidants play an important role in the prevention and treatment of diseases, and their development has attracted increasing attention [24,25,26]. Knowing that polyhydroxy compounds are potent antioxidants, and that they are widely present in plants [27,28], this study investigates the polyhydroxy (two or three hydroxy groups) indane derivatives in *A. tanguticus*. In total, four novel compounds are isolated, and their antioxidant activities are evaluated using ABTS^•+^ and DPPH^•+^ assays. The obtained results reveal that all four indane derivatives (**1**–**4**) identified herein have good free radial scavenger activities, with **1** being the most active compound. Comparison of the structures of compounds **1**–**4** suggests that the strong activity of compound **1** may be attributed to the OH-1′ substitution and the configuration of C-2′. However, ABTS^•+^ and DPPH^•+^ assays are just simplified methods for estimating antioxidant activity, which do not reflect the actual antioxidant activity [29]. Unfortunately, further studies on the antioxidant activity of compounds **1**–**4** could not be carried out due to their limited sample quantities.

Indene analogs generally have significant neuroprotective effects [21]. Indeed, donepezil, a second-generation AChE inhibitor, is used to treat Alzheimer’s disease due to its significant cholinesterase inhibitory activity. In light of this information, as well as the strong AChE inhibitory effect of the *A. tanguticus* extract [30], the AChE inhibitory activity of the compounds isolated herein is also analyzed in this study using the modified Ellman method [31] and donepezil as a positive control. However, none of the compounds show inhibitory activity against AChE, even at concentrations as high as 100 μM.

## 4. Materials and Methods

### 4.1. General Experimental Procedures

Optical rotation was measured using a Rudolph Autopol I automatic polarimeter (Rudolph Research Analytical, Hackettstown, NJ, USA). ECD and IR spectra were recorded on an Applied Photophysics Chirascan CD spectrometer (Applied Photophysics Ltd., Leatherhead, UK) and an Agilent Cary 600 FT-IR microscope instrument (Agilent Technologies Inc., Santa Clara, CA, USA), respectively. X-ray crystallographic analyses were performed on a Bruker D8 Quest diffractometer (Bruker Corporation, Billerica, MA, USA). Meanwhile, NMR and HR-ESI-MS spectra were acquired on a Bruker Avance NEO 600 or a Bruker Avance NEO 700 spectrometer (solvent peaks used as the references) and a Waters Synapt G2 HDMS (Waters Corporation, Milford, MA, USA) or a Bruker timsTOF MS instrument, respectively. The melting points were measured on a BÜCHI M-565 melting point apparatus (BÜCHI Labortechnik AG, Flawil, Switzerland). MPLC separations were carried out using a BÜCHI Pure C-805 instrument. HPLC separations were performed on an Agilent 1220 instrument with a Welch Ultimate XB-C_18_ column (10 × 250 mm^2^, 5 μm) or a Daicel Chiralpak AD-H column (4.6 × 250 mm^2^, 5 μm). TLC was carried out using silica gel GF254 plates (Anhui Liangchen Silicon Material Co. Ltd., Lu’an, Anhui, China), whereas column chromatography separations were performed on silica gel (200–300 mesh, Yantai Institute of Chemical Technology, Yantai, Shandong, China), Sephadex LH-20 (40–70 μm, Amersham Pharmacia Biotech AB, Uppsala, Sweden), or ODS (40 μm, Acchrom Technology Co. Ltd., Beijing, China).

### 4.2. Plant Material

The roots of *A. tanguticus* (Maxim.) Pascher (Solanaceae) were collected from Aba Tibetan and Qiang Autonomous Prefecture in Sichuan Province, China, during the month of October in 2017. The collected samples were identified by Dr. Ji-hai Gao (Chengdu University of TCM, Chengdu, Sichuan, China) and deposited at Chengdu No. 1 Pharmaceutical Co. Ltd. in Chengdu, Sichuan, China.

### 4.3. Extraction and Isolation

The powdered roots of *A. tanguticus* (500 kg) were dampened with ammonia, and then extracted with diethoxymethane (6 × 500 L) under countercurrent extraction for 2 h at room temperature. The extracting solution was partitioned with 20% sulfuric acid, affording aqueous and organic phases. The organic phase was concentrated under reduced pressure to yield a residue (3 kg). Part of the residue (1 kg) was suspended in H_2_O and successively partitioned with petroleum ether and EtOAc. The EtOAc fraction (60 g) was subjected to silica gel column chromatography and eluted with petroleum ether–EtOAc (50:1–1:1) and EtOAc–MeOH (1:0–0:1) to afford 9 fractions (Fr.1–Fr.9). Among them, Fr.7 (17.9 g) was chromatographed over a Sephadex LH-20 column (petroleum ether–CH_2_Cl_2_–MeOH, 5:5:1) to give 12 subfractions (Fr.7-1–Fr.7-12). Subfraction Fr.7-11 (2.8 g) was separated into six subfractions (Fr.7-11-1–Fr.7-11-6) on a Sephadex LH-20 column (MeOH–H_2_O, 80:20). Subsequently, subfraction Fr.7-11-4 (153 mg) was separated by silica gel chromatography (CH_2_Cl_2_–Me_2_CO, 20:1–1:1) and reverse-phase semi-preparative HPLC (40% MeOH in H_2_O) into several mixtures, including a mixture of compound **3** and **4** (9 mg, t_R_ = 28.7 min). The pure compounds were chirally separated on a Daicel Chiralpak AD-H column using normal-phase HPLC (*n*-hexane/ethanol, 5:1). Compound **3** (4 mg) was eluted at 9.5 min, whereas compound **4** (4 mg) was eluted at 14.3 min. Fr.7-12 (500 mg) was first separated by silica gel column chromatography into five subfractions (Fr.7-12-1–Fr.7-12-5), using CH_2_Cl_2_–MeOH (100:1–10:1) as the mobile phase. Thereafter, Fr.7-12-1 (150 mg) was purified by preparative TLC (CH_2_Cl_2_–MeOH, 17:1) and reverse-phase semi-preparative HPLC (50% MeOH in H_2_O), followed by normal-phase HPLC (*n*-hexane/ethanol, 3:1) to afford compounds **1** (5 mg, t_R_ = 11.8 min) and **2** (2 mg, t_R_ = 6.5 min).

### 4.4. Physicochemical Properties and Spectroscopic Data of Compounds ***1**–**4***

Anisotindan A [(2*R*)-5-Hydroxy-4-methyl-2-((2*S*)-1,2-dihydroxyisopropyl)indane] (**1**): colorless crystals; mp 174–176 °C; [*α*${]}_{D}^{20}$ −17.0 (*c* 0.02, MeOH); UV (MeCN) *λ*_max_ (log *ε*) 282 (2.89), 218 (3.57), 199 (4.32) nm; ECD (MeCN) 194 (Δ*ε* −3.15), 283 (Δ*ε* −0.24) nm; IR *ν*_max_ 3358, 2922, 2852, 1658, 1634, 1604, 1541, 1470, 1384, 1263, 1049, 1031, 940, 862, 807 cm^−1^; ^1^H NMR (acetone-*d*_6_, 600 MHz) and ^13^C NMR (acetone-*d*_6_, 150 MHz) data, see Table 1; (+)-HR-ESI-MS *m/z* 245.1157 [M + Na]^+^ (calcd. for C_13_H_18_O_3_Na, 245.1154). The original UV, IR, (+)-HR-ESI-MS, ^1^H NMR, ^13^C NMR, DEPT, HSQC, ^1^H-^1^H COSY, and HMBC spectra are shown in Appendix A.

Anisotindan B [(2*R*)-5-Hydroxy-4-methyl-2-((2*R*)-1,2-dihydroxyisopropyl)indane] (**2**): white powder; [*α*${]}_{D}^{20}$ −15.0 (*c* 0.02, MeOH); UV (MeCN) *λ*_max_ (log *ε*) 283 (2.96), 218 (3.65), 199 (4.47) nm; ECD (MeCN) 197 (Δ*ε* −4.58), 282 (Δ*ε* −0.50) nm; IR *ν*_max_ 3424, 2935, 2850, 1659, 1631, 1603, 1455, 1349, 1266, 1160, 1071, 1027, 940, 811 cm^−1^; ^1^H NMR (acetone-*d*_6_, 600 MHz) and ^13^C NMR (acetone-*d*_6_, 150 MHz) data, see Table 1; (+)-HR-ESI-MS *m/z* 245.1148 [M + Na]^+^ (calcd. for C_13_H_18_O_3_Na, 245.1154). The original UV, IR, (+)-HR-ESI-MS, ^1^H NMR, ^13^C NMR, DEPT, HSQC, ^1^H-^1^H COSY, and HMBC spectra are shown in Appendix A.

Anisotindan C [(3*R*,3a*S*,8b*S*)-3,6-Dihydroxy-3,5-dimethyl-3,3a,4,8b-tetrahydro-2*H*-indeno[1,2-*b*]furan] (**3**): colorless crystals; mp 209–211 °C; [*α*${]}_{D}^{20}$ +19.0 (*c* 0.03, MeOH); UV (MeCN) *λ*_max_ (log *ε*) 279 (2.83), 221 (3.70), 200 (4.54) nm; ECD (MeCN) 191 (Δ*ε* −4.19), 204 (Δ*ε* 0.55), 209 (Δ*ε* −0.97), 230 (Δ*ε* 2.49) nm; IR *ν*_max_ 3357, 3278, 2921, 2850, 1659, 1633, 1603, 1470, 1426, 1383, 1353, 1271, 1243, 1152, 1132, 1024, 970, 934, 893, 816, 754, 705 cm^−1^; ^1^H NMR (acetone-*d*_6_, 700 MHz) and ^13^C NMR (acetone-*d*_6_, 175 MHz) data, see Table 1; (+)-HR-ESI-MS *m/z* 243.0996 [M + Na]^+^ (calcd. for C_13_H_16_O_3_Na, 243.0997). The original UV, IR, (+)-HR-ESI-MS, ^1^H NMR, ^13^C NMR, DEPT, HSQC, ^1^H-^1^H COSY, HMBC, and 1D NOE spectra are shown in Appendix A.

Anisotindan D [(3*S*,3a*R*,8b*R*)-3,7-Dihydroxy-3,8-dimethyl-3,3a,4,8b-tetrahydro-2*H*-indeno[1,2-*b*]furan] (**4**): white powder; [*α*${]}_{D}^{20}$ −53.0 (*c* 0.05, MeOH); UV (MeCN) *λ*_max_ (log *ε*) 285 (3.28), 218 (3.86), 199 (4.58) nm; ECD (MeCN) 193 (Δε 2.32), 207 (Δ*ε* −7.37), 231 (Δ*ε* 2.28) nm; IR *ν*_max_ 3356, 3270, 2921, 2851, 1659, 1633, 1498, 1469, 1429, 1382, 1333, 1266, 1180, 1051, 1023, 974, 919, 882, 815, 717, 707 cm^−1^; ^1^H NMR (acetone-*d*_6_, 700 MHz) and ^13^C NMR (acetone-*d*_6_, 175 MHz) data, see Table 1; (+)-HR-ESI-MS *m/z* 243.0997 [M + Na]^+^ (calcd. for C_13_H_16_O_3_Na, 243.0997). The original UV, IR, (+)-HR-ESI-MS, ^1^H NMR, ^13^C NMR, DEPT, HSQC, ^1^H-^1^H COSY, HMBC, and 1D NOE spectra are shown in Appendix A.

### 4.5. X-ray Crystallographic Data of Compounds ***1*** and ***3***

Crystals of **1** and **3** were obtained from MeOH. Intensity data were collected on a Bruker D8 Quest diffractometer equipped with an APEX-II CCD using Cu K*α* radiation. Crystallographic data for the reported structures have been deposited at the Cambridge Crystallographic Data Centre (CCDC). Copies of the data can be acquired free of charge from CCDC, 12 Union Road, Cambridge CB2 1EZ, UK (fax: +44-1223-336-033; e-mail: deposit@ccdc.cam.ac.uk).

Crystal data for **1**: C_13_H_18_O_3_, M = 222.27, colorless crystals, monoclinic, *a* = 6.7889(2) Å, *b* = 8.5863(2) Å, *c* = 10.5065(3) Å, *a* = 90°, *β* = 106.502(1)°, *γ* = 90°, *V* = 587.21(3) Å^3^, space group *P*2_1_, *T* = 293(2) K, *Z* = 2, *μ*(Cu K*α*) = 0.713 mm^−1^, 10,969 reflections measured, 2125 independent reflections (*R*_int_ = 0.0395). Final *R* indices (*I* > 2*σ*(*I*)): *R*_1_ = 0.0344, *wR*_2_ = 0.0810. Final *R* indices (all data): *R*_1_ = 0.0385, *wR*_2_ = 0.0844. The goodness of fit on *F*^2^ was 1.093. Flack parameter = 0.02(13). CCDC number: 2232633.

Crystal data for **3**: C_13_H_16_O_3_, M = 220.26, colorless crystals, orthorhombic, *a* = 6.156(3) Å, *b* = 9.879(4) Å, *c* = 18.688(9) Å, *a* = 90°, *β* = 90°, *γ* = 90°, *V* = 1136.4(9) Å^3^, space group *P*2_1_2_1_2_1_, *T* = 273(2) K, *Z* = 4, *μ*(Cu K*α*) = 0.736 mm^−1^, 41,583 reflections measured, 2086 independent reflections (*R*_int_ = 0.0513). Final *R* indices (*I* > 2*σ*(*I*)): *R*_1_ = 0.0347, *wR*_2_ = 0.0924. Final *R* indices (all data): *R*_1_ = 0.0367, *wR*_2_ = 0.0951. The goodness of fit on *F*^2^ was 1.056. Flack parameter = 0.08(7). CCDC number: 2232632.

### 4.6. ECD Calculation

The details of ECD calculation of compounds **2** and **4** are shown in Appendix A.

### 4.7. Antioxidant Activity

The ABTS and DPPH free radical scavenging assays were used to estimate the antioxidant activities of the isolated compounds.

#### 4.7.1. ABTS^•+^ Assay

The free radical scavenging capacity of compounds **1**–**4** was measured using the ABTS^•+^ decoloration method. First, 20 of mL ABTS^•+^ solution (7 mM) and 20 mL of potassium persulfate solution (2.45 mM) were prepared with ultra-pure water. The prepared solutions were mixed and stored in the dark at 23 °C for 16 h to obtain ABTS^•+^ stock solution. Two milliliters of this solution were subsequently diluted (20 times) with 95% ethanol solution to obtain the ABTS^•+^ working solution with an absorbance of 0.70 ± 0.02 at 734 nm. Thereafter, compound solutions (80 μL) of varying concentrations were mixed with 400 μL of the ABTS^•+^ working solution and added into 96-well plates, with 150 μL in each well. After 6 min incubation in the dark at 23 °C, the absorbance (OD) of each sample was measured at 734 nm. Using vitamin C as the positive control and 95% ethanol solution as the blank control, the scavenging rate of ABTS free radicals was calculated according to the following equation: (%) = (1 − *A*_s_/*A*_c_) × 100%, where *A*_s_ and *A*_c_ are the average OD values of the drug group and the blank control group, respectively. All tests were performed in triplicate.

#### 4.7.2. DPPH^•+^ Assay

DPPH^•+^ solution (0.1 mM) was prepared with 95% ethanol, and 250 μL of the solution was mixed with compound solutions (250 μL) of varying concentrations. The mixtures were transferred into 96-well plates, with 150 μL in each well. After the reaction at 25 °C for 30 min, the absorbance (OD) was measured at 517 nm, and the DPPH free radical scavenging rate was calculated according to the following equation: (%) = (1 − *A*_s_/*A*_c_) × 100%, where *A*_s_ and *A*_c_ are the average OD values of the drug group and the blank control group (95% ethanol solution), respectively. All tests were performed in triplicate.

## 5. Conclusions

In this study, four new indanes (**1**–**4**), anisotindans A–D, were extracted from the roots of *A. tanguticus*. Their structures were identified by NMR and single-crystal X-ray crystallography analyses, as well as ECD calculations. Meanwhile, their antioxidant activity was estimated using ABTS and DPPH free radical scavenging assays. The obtained results show that anisotindans C and D (**3** and **4**) are two unusual indenofuran analogs, and that compounds **1**–**4** exhibit significant antioxidant capacity, especially compound **1**, the ABTS radical scavenging capacity of which is greater than that of vitamin C. The preliminary structure–activity relationship analysis conducted herein suggests that the variation in antioxidant activity of indanes may be attributed to differences in OH-1′ substitution and C-2′ configuration.

## Figures and Tables

**Figure 1 molecules-28-01493-f001:**
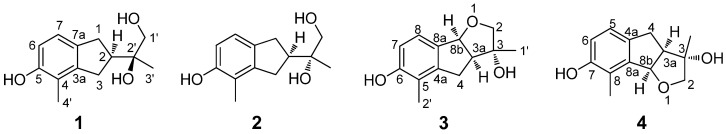
Structures of the novel indanes (**1**–**4**) isolated from *A. tanguticus*.

**Figure 2 molecules-28-01493-f002:**
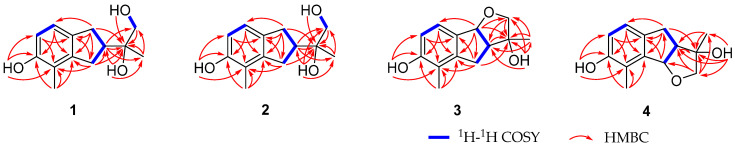
Key ^1^H-^1^H COSY and HMBC correlations of compounds **1**–**4**.

**Figure 3 molecules-28-01493-f003:**
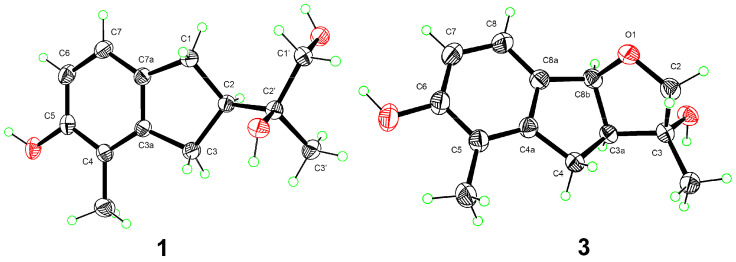
X-ray crystallographic structures of compounds **1** and **3**.

**Figure 4 molecules-28-01493-f004:**
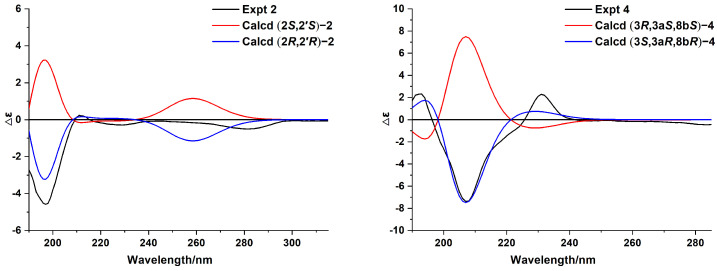
Experimental and calculated ECD spectra of compounds **2** and **4**.

**Table 1 molecules-28-01493-t001:** ^1^H and ^13^C NMR data for compounds **1**–**4** in acetone-*d*_6_ (*δ* in ppm, *J* in Hz).

No.	1 *^a^*		2 *^a^*		3 *^b^*		4 *^b^*	
	*δ* _H_	*δ* _C_	*δ* _H_	*δ* _C_	*δ* _H_	*δ* _C_	*δ* _H_	*δ* _C_
1	2.91 dd (15.0, 9.0) 2.76 dd (15.0, 9.0)	33.5	2.82 dd (15.0, 9.0) 2.78 dd (15.0, 9.0)	34.1				
2	2.65 m	47.5	2.66 m	47.5	3.58 dd (8.4, 0.7) 3.36 d (8.4)	77.9	3.61 dd (9.1, 0.7) 3.37 d (9.1)	78.1
3	2.82 m	33.4	2.84 overlapped	32.8		80.5		80.4
3a		144.3		144.7	2.91 m	55.5	2.91 overlapped	55.4
4		120.4		120.5	2.95 dd (15.4, 9.1) 2.73 dd (15.4, 4.2)	33.3	2.91 overlapped 2.76 dd (20.3, 9.8)	33.8
4a						145.1		134.7
5		154.4		154.4		120.1	6.81 d (7.7)	122.5
6	6.59 d (8.4)	113.6	6.59 d (8.4)	113.6		156.5	6.74 d (7.7)	116.6
7	6.77 d (8.4)	122.2	6.78 d (8.4)	122.2	6.70 d (8.4)	114.7		155.0
7a		134.7		134.4				
8					6.97 d (8.4)	123.9		122.3
8a						134.1		143.3
8b					5.47 d (6.3)	88.1	5.61 d (6.3)	87.2
1′	3.48 dd (10.8, 6.0) 3.45 dd (10.8, 6.0)	69.8	3.49 dd (10.2, 6.0) 3.46 dd (10.2, 6.0)	69.9	1.32 s	20.9	1.30 s	20.7
2′		73.7		73.7	2.09 s	12.1	2.22 s	12.0
3′	1.19 s	22.9	1.17 s	22.8				
4′	2.09 s	12.4	2.09 s	12.4				
OH-3					3.83 s		3.87 s	
OH-5	7.77 s		7.76 s					
OH-6					8.10 s			
OH-7							7.92 s	
OH-1′	3.70 t (6.0)		3.71 t (6.0)					
OH-2′	3.28 s		3.29 s					

*^a^* Data were measured at 600 MHz for ^1^H and 150 MHz for ^13^C. *^b^* Data were measured at 700 MHz for ^1^H and 175 MHz for ^13^C.

**Table 2 molecules-28-01493-t002:** ABTS^•+^ and DPPH^•+^ scavenging activities of compounds **1**–**4**
^a^.

	IC_50_ for ABTS^•+^ Scavenging Assay (μM)	IC_50_ for DPPH^•+^ Scavenging Assay (μM)
**1**	15.62 ± 1.85	68.46 ± 17.34
**2**	40.92 ± 7.02	>100
**3**	43.93 ± 9.35	>100
**4**	32.38 ± 6.29	>100
VC	22.54 ± 5.18	10.19 ± 1.38

^a^ All values are represented as Mean ± SD, *n* = 3.

## Data Availability

The data presented in this study are available in the Appendix A or can be provided by the authors.

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
