# Peer review of "Novel Indane Derivatives with Antioxidant Activity from the Roots of Anisodus tanguticus"

_molecules, 2023, doi:10.3390/molecules28031493_

Round 1
Reviewer 1 Report
The manuscript reports the isolation of some novel indane derivatives having a phenolic moiety. They are therefore tested for the antioxidant activity by the DPPH and ABTS tests. The manuscript is interesting and the research is carefully performed. However, the part about the antioxidant activity must be improved before publication.
1) the results reported by the authors on the order of radical trapping of compounds 1-4 is incredible. Why 1 is active, whereas 2-4 are not? The reactivity of phenols with radicals depends on the groups attached to the phenolic ring, not on the geometry of some "remote" substituents that have no interaction with the aromatic ring. (see DOI: 10.1039/b901838g) Probably the reason for this unexpected result is that DPPH and ABTS represent simplified methods to measure the antioxidant activity. The procedure used by the Authors is actually a "titration" of the reducing activity of 1-4, that does not correlate necessarily with the rate constant of reaction of 1-4 with radicals (see for example https://doi.org/10.3109/10715762.2014.996146). May be that the oxidized form of 1 can undergo some unique reaction that requires a specific geometry. It would be interesting to know what if formed after the reaction of 1-4 with these stable radicals. Anyway, the Authors should clearly state that DPPH and ABTS provide only an estimate. In facts, they are different from peroxyl radicals (ROO*) that propagate lipid peroxidation, so the results obtained with them may not reflect the actual activity.
2) If the authors have the possibility, they should try a method based on lipid peroxidation, for example by oxidizing methyl linoleate and measuring the formation of conjugated dienes.
3) As a consequence, The following sentence is wrong. "Knowing that the antioxidant activity of indane compounds depends on their absolute configuration, as well as the number of hydroxy groups [28–31], the strong activity of compound 1 may be attributed to the OH-9 substitution and the configuration of C-8."
First, they did not measure an antioxidant activity, but a radical trapping ability.
Second, the number of OH groups refers only to the number of phenolic OH. Also, as previously explained, it is unclear why configuration at C8 should influence activity.
Minor points:
"compound 3 is a 1,9-epoxy derivative of compound 2" Are Authors sure that "epoxy" is the right definition for 3? Epoxy is used for 3-membered rings.
in the optical rotation, it appears that the Authors forgot = and the unit of the concentration.
I would move the crystallographic data from the results to the experimental section.
In the section: Details on the extraction and preliminary separation procedures are available in the references [10,11]. I would summarise all the procedure of extraction.
", the scavenging rate of ABTS free radicals was calculated..." "the DPPH free radical scavenging rate was calculated" These sentences are wrong. The procedure reported does not allow to measure a reaction rate, but the amount of radical trapped after a fixed time.
I would add some recent citations about the chemistry of antioxidants. For example: DOI: 10.1039/d1cs00265a
Author Response
Comment 1
The results reported by the authors on the order of radical trapping of compounds 1-4 is incredible. Why 1 is active, whereas 2-4 are not? The reactivity of phenols with radicals depends on the groups attached to the phenolic ring, not on the geometry of some "remote" substituents that have no interaction with the aromatic ring. (see DOI: 10.1039/b901838g). Probably the reason for this unexpected result is that DPPH and ABTS represent simplified methods to measure the antioxidant activity. The procedure used by the Authors is actually a "titration" of the reducing activity of 1-4, that does not correlate necessarily with the rate constant of reaction of 1-4 with radicals (see for example https://doi.org/10.3109/10715762.2014.996146). May be that the oxidized form of 1 can undergo some unique reaction that requires a specific geometry. It would be interesting to know what if formed after the reaction of 1-4 with these stable radicals. Anyway, the Authors should clearly state that DPPH and ABTS provide only an estimate. In facts, they are different from peroxyl radicals (ROO*) that propagate lipid peroxidation, so the results obtained with them may not reflect the actual activity.
Response
Thank you very much for your professional comment and valuable advice. We are in complete agreement with your opinions. We have revised the relevant description in the “Discussion”.
Comment 2
If the authors have the possibility, they should try a method based on lipid peroxidation, for example by oxidizing methyl linoleate and measuring the formation of conjugated dienes.
Response
Special thanks for this helpful suggestion. However, the quantities of these natural products isolated from Anisodus tanguticus are small. In addition, the chemical studies and activity assays have used most of the samples. Thus, we regret that we are unable to further investigate the inhibitory activity against lipid peroxidation due to the limited sample quantities of the isolates. Thanks again for your good comment.
Comment 3
As a consequence, the following sentence is wrong. "Knowing that the antioxidant activity of indane compounds depends on their absolute configuration, as well as the number of hydroxy groups [28–31], the strong activity of compound 1 may be attributed to the OH-9 substitution and the configuration of C-8."
First, they did not measure an antioxidant activity, but a radical trapping ability.
Second, the number of OH groups refers only to the number of phenolic OH. Also, as previously explained, it is unclear why configuration at C8 should influence activity.
Response
We are appreciative of your valuable comment. We have revised the relevant description in the “Discussion”.
Comment 4
"compound 3 is a 1,9-epoxy derivative of compound 2" Are Authors sure that "epoxy" is the right definition for 3? Epoxy is used for 3-membered rings.
Response
Thanks for your comment. We think that "epoxy" is a suitable definition for compounds 3 and 4. In IUPAC, “epoxy” is not only used for 3-membered rings. For example, the oxygen rings in the 7,7'-epoxylignanes, 2',7-epoxy-6'H-8,1'-neolignanes, 7,9':7',9-diepoxylignanes, and 2',7-epoxy-8,1'-neolignanes are 5-membered; the oxygen rings in the 3',7-epoxy-8,4'-oxyneolignanes are 6-membered (Pure Appl. Chem., 2000, 72(8): 1493–1523). To consider this suggestion, we have revised “1,9-epoxy derivative” as “1-O-9 ether derivative” in the manuscript.
Comment 5
In the optical rotation, it appears that the Authors forgot = and the unit of the concentration.
Response
Thanks for your suggestion. Generally, the "=" symbol and the default concentration unit (g/100ml) should be omitted when reporting optical rotation data of natural compounds. This format is very common in most related journals and is also adopted by Molecules.
Comment 6
I would move the crystallographic data from the results to the experimental section.
Response
Thanks for your helpful suggestion. We have moved the physicochemical properties, spectroscopic data, and crystallographic data from the results to the experimental section.
Comment 7
In the section: Details on the extraction and preliminary separation procedures are available in the references [10,11]. I would summarise all the procedure of extraction.
Response
Thanks for your helpful suggestion. The details on the extraction and preliminary separation procedures have been added in section 4.3.
Comment 8
", the scavenging rate of ABTS free radicals was calculated..." "the DPPH free radical scavenging rate was calculated" These sentences are wrong. The procedure reported does not allow to measure a reaction rate, but the amount of radical trapped after a fixed time.
Response
Thanks for your comment. This “rate” does not mean the reaction rate, but refer to the scavenging ratio of ABTS free radicals.
Comment 9
I would add some recent citations about the chemistry of antioxidants. For example: DOI: 10.1039/d1cs00265a.
Response
Thanks for your suggestion. We have added this reference.
Reviewer 2 Report
Dear Editor and Authors,
The manuscript ‘Novel indane derivatives with antioxidant activity from the roots of Anisodus tanguticus’ by Chun-Wang Meng, Hao-Yu Zhao , Huan Zhu, Cheng Peng, Qin-Mei Zhou, and Liang Xiong describes research on identification of 4 novel indane derivatives, anisotindans A–D (1–4), were isolated from the roots of Anisodus tanguticus. The Authors used sophisticated spectroscopic analyses, determinated the absolute configurations by electronic circular dichroism (ECD) calculations and single-crystal X-ray diffraction analyses. They used ABTS•+ and DPPH•+ assays of radical scavenging activity to measure the antioxidant activities of the compounds. The manuscript is well written, and the methods used are sophisticated.
I only do not like the conclusion that all the compounds express good antioxidant activities.
The Authors concluded:
‘The obtained results reveal that all four indane derivatives (1–4) identified herein have good antioxidant activities..’ (Page 6)
Based on Table 2 we can see that compound 1 has lower antioxidant activities than vitamin C (by ABTS•+ scavenging assay, μM), compounds 2-4 up to twice the activity of vitamin C, while by IC50 for DPPH•+ scavenging assay (μM) test only compound 1 showed some activity. I would not strongly press the antioxidant activity of the compounds. I would suggest rewriting the sentence to be more accurate.
That is why, I recommend minor revision of the manuscript,
Yours sincerely,
Author Response
Comment
The manuscript ‘Novel indane derivatives with antioxidant activity from the roots of Anisodus tanguticus’ by Chun-Wang Meng, Hao-Yu Zhao, Huan Zhu, Cheng Peng, Qin-Mei Zhou, and Liang Xiong describes research on identification of 4 novel indane derivatives, anisotindans A–D (1–4), were isolated from the roots of Anisodus tanguticus. The Authors used sophisticated spectroscopic analyses, determinated the absolute configurations by electronic circular dichroism (ECD) calculations and single-crystal X-ray diffraction analyses. They used ABTS•+ and DPPH•+ assays of radical scavenging activity to measure the antioxidant activities of the compounds. The manuscript is well written, and the methods used are sophisticated. I only do not like the conclusion that all the compounds express good antioxidant activities.
The Authors concluded: ‘The obtained results reveal that all four indane derivatives (1–4) identified herein have good antioxidant activities..’ (Page 6)
Based on Table 2 we can see that compound 1 has lower antioxidant activities than vitamin C (by ABTS•+ scavenging assay, μM), compounds 2-4 up to twice the activity of vitamin C, while by IC50 for DPPH•+ scavenging assay (μM) test only compound 1 showed some activity. I would not strongly press the antioxidant activity of the compounds. I would suggest rewriting the sentence to be more accurate.
Response
Thank you for your comment. Table 2 shows the IC50 (half maximal inhibitory concentration) values of compounds 1-4. The lower the IC50 value, the better the effect. The IC50 values of compounds 1-4 and vitamin C are 15.62 ± 1.85, 40.92 ± 7.02, 43.93 ± 9.35, 32.38 ± 6.29, and 22.54 ± 5.18 μM, respectively. Thus, we described that “The activity of compound 1, the most potent scavenger, is even stronger than that of vitamin C.”
Reviewer 3 Report
The crystal structures of compounds 1 and 3 are nicely determined, including unequivocal determination of their absolute configurations, agreeing with chiroptical data. However, there are a few problems with the CIFs, which need to be cleaned up:
For compound 1, there are some minor differences between values given in the manuscript and those in the CIF. These should be made consistent.
For both compounds, the atom numbering in the CIFs is random and does not agree with that shown in Figures 1 and 3. Were preliminary models deposited at the CSD before the numbering in Figure 3 was adopted?
For both compounds, the dimensions of the crystal used are missing from the CIFs.
Information about absorption corrections are missing from both CIFs.
For both refinements, the SHELXL weight optimization had not quite converged.
These issues should be attended to, and replacement CIFs should be deposited with the CCDC.
Author Response
Comment 1
For compound 1, there are some minor differences between values given in the manuscript and those in the CIF. These should be made consistent.
Response
Thank you very much for your comment. We have corrected minor errors.
Comment 2
For both compounds, the atom numbering in the CIFs is random and does not agree with that shown in Figures 1 and 3. Were preliminary models deposited at the CSD before the numbering in Figure 3 was adopted?
Response
Thank you very much for your valuable advice. The atom numbering in the CIFs has been revised.
Comment 3
For both compounds, the dimensions of the crystal used are missing from the CIFs. Information about absorption corrections are missing from both CIFs.
Response
Thanks for your comment. We have added the dimensions of the crystal and the information about absorption corrections in the CIFs.
Comment 4
For both refinements, the SHELXL weight optimization had not quite converged.
Response
Thank you for your good suggestion. We have tried our best to refine the SHELXL weight optimization. Although it had not quite converged, the current data are sufficient to confirm the accuracy of the structure.
Comment 5
These issues should be attended to, and replacement CIFs should be deposited with the CCDC.
Response
Thanks for your valuable advice. The modified CIFs have been updated in the CCDC.
Reviewer 4 Report
The presented manuscript describes the isolation, identification and antioxidant activity of 4 new indane-type compounds from the plant Anisodus tanguticus. To the best of my knowledge the compounds have not been previously described. The authors used proper analytical and physico-chemical methods in the characterization process of the compounds. The possibility to determine the absolute configuration of the compounds (especially the acyclic fragments in 1 and 2 is appreciated. The manuscript should definitely be published; the paper might not attract the attention of a wide readership of the journal, but its inclusion in the special issue is a good intention.
There are only very few negative aspects that can be found; one of them being the fact that, if I understood correctly, two papers have already been published by the authors about the isolation of compounds from the same starting plant material and its extract (but from different chromatographic fractions) - I fully understand the difficulty and lengthiness of the isolation and identification process (reflected in the varied years of publication of the data - 2020, 2021 and now), but the impact of the whole study might be diminished by splitting it into several small fragments ('salami publications' as they are sometimes referred to as). No specific action needs to be taken by the authors at the moment regarding this issue though.
Please, add 'Hz' after the coupling constant values in section 2.1., lines 5 and 6.
Remove bold formatting of the second parenthesis in 'Asotindan B (2)' below Table 1.
I suggest replacing 'was agreed with the experimental ECD spectrum...' with 'was in agreement with the experimental ECD spectrum...' at the end of the text in sections ECD Calculation of Compound 2 and ECD Calculation of Compound 4 - both in the supporting information file.
IUPAC suggests writing a space between the temperature value and the °C unit.
I suggest writing symbols of quantities in the Supporting Information file in italics (calculation sections, tables S1 and S2).
In the sentence 'Knowing that the antioxidant activity of indane compounds depends on their absolute configuration, as well as the number of hydroxy groups [28–31], the strong activity of compound 1 may be attributed to the OH-9 substitution and the configuration of C-8.' in the second paragraph of section 3. the references do not explicitly define the dependence of the antioxidant activity of indane compounds on their absolute configuration and the number of hydroxy groups. Indeed, the references deal with the varied antioxidant activities of cis- and trans- isomers of certain compounds, and also the its variation with the amount and substitution of -OH groups of different types of compounds. I would say that the dependence in the case of indane compounds is not established, but its existence can be presumed from the literature and from the authors' results.
The author guidelines of the journal suggests using IUPAC nomenclature (or e. g. CAS rules). The authors coin the new names Anisotindan A-D which is in accordance with the general practice in natural chemistry research; I would suggest adding a non-trivial name next to the individual trivial names in section 2.2. for easier searching in literature without structure search engines.
Congratulations.
Author Response
Comment 1
Please, add 'Hz' after the coupling constant values in section 2.1., lines 5 and 6. Remove bold formatting of the second parenthesis in 'Asotindan B (2)' below Table 1.
Response
Thanks for your careful examination of our manuscript. All format errors have been corrected in the revised manuscript.
Comment 2
I suggest replacing 'was agreed with the experimental ECD spectrum...' with 'was in agreement with the experimental ECD spectrum...' at the end of the text in sections ECD Calculation of Compound 2 and ECD Calculation of Compound 4 - both in the supporting information file. IUPAC suggests writing a space between the temperature value and the °C unit. I suggest writing symbols of quantities in the Supporting Information file in italics (calculation sections, tables S1 and S2).
Response
Thank you for your helpful suggestions. We have revised these descriptions.
Comment 3
In the sentence 'Knowing that the antioxidant activity of indane compounds depends on their absolute configuration, as well as the number of hydroxy groups [28–31], the strong activity of compound 1 may be attributed to the OH-9 substitution and the configuration of C-8.' in the second paragraph of section 3. the references do not explicitly define the dependence of the antioxidant activity of indane compounds on their absolute configuration and the number of hydroxy groups. Indeed, the references deal with the varied antioxidant activities of cis- and trans- isomers of certain compounds, and also the its variation with the amount and substitution of -OH groups of different types of compounds. I would say that the dependence in the case of indane compounds is not established, but its existence can be presumed from the literature and from the authors' results.
Response
We are appreciative of your valuable comment. We have revised the relevant description in the “Discussion”.
Comment 4
The author guidelines of the journal suggests using IUPAC nomenclature (or e. g. CAS rules). The authors coin the new names Anisotindan A-D which is in accordance with the general practice in natural chemistry research; I would suggest adding a non-trivial name next to the individual trivial names in section 2.2. for easier searching in literature without structure search engines.
Response
We are appreciative of your valuable advice. We have added the non-trivial names of compounds 1-4 next to the individual trivial names in section 2.2. In addition, the atom numbering has also been revised in the structures, texts, and Tables.
Round 2
Reviewer 1 Report
The authors addressed my concerns.
However, in the yellow part of the discussion there are some spelling errors. Also, it is unclear what is intended with "9-OH" group.